# The Future of Foundation Models in Predicting Climate-Related Risks in the Insurance Sector: A Case Study in Louisiana

Meghna Behari
Harvard College
Cambridge, Massachussetts
meghnabehari@college.harvard.edu

## ABSTRACT

Climate change continues to pose a significant threat to the world, contributing to the increase in frequency and severity of natural disasters. This escalating risk directly impacts the U.S. homeowners' insurance industry, leading to higher premiums and reduced coverage availability in vulnerable areas of the country. Traditional methods of modeling insurance premiums often fail to fully account for the multifaceted effects of climate risk, insurer risk assessments, and the spatial-temporal variance of climate impacts. This study focuses on Louisiana, a state particularly vulnerable to climate change, to illustrate these gaps. We propose incorporating foundation models into climate risk assessment and insurance modeling to gain deeper insights into the impact of climate change on insurance premiums and the disparities in equal access to affordable insurance. This approach supports greater climate resilience aligning with the UN Sustainable Development Goal 13 to strengthen adaptive capacity to climate hazards.

## KEYWORDS

Supervised learning, insurance, climate change, foundation models

**ACM Reference Format:**
Meghna Behari. 2024. The Future of Foundation Models in Predicting Climate-Related Risks in the Insurance Sector: A Case Study in Louisiana. In . ACM, New York, NY, USA, 5 pages. https://doi.org/XXXXXXX.XXXXXXX

## 1 INTRODUCTION

Climate change continues to have profound impacts across the globe, increasing the incidences of catastrophic climate-linked disasters. These climatic changes pose a threat to human health, but are also associated with significant economic risks. As the climate crisis intensifies, it becomes increasingly crucial to understand and mitigate its economic consequences.

The U.S. homeowners' insurance industry is particularly vulnerable to climate change. Increased climate-related destruction has, in recent years, led to higher insurance rates, making it challenging for insurers to provide affordable coverage [1]. In some regions, the escalating risk has even prompted major insurers to withdraw from the market entirely [2], reducing coverage availability and leaving homeowners unprotected. This dynamic is especially pronounced in high-risk areas such as the state of Louisiana, which is prone to hurricanes and coastal flooding.

Traditional actuarial models used in the insurance industry primarily rely on historical data and statistical methods to predict future risks and set premiums[3]. While these models have been effective to an extent, they fall short in capturing the dynamics of long-term climate-related risks [4]. Basic machine learning models offer improvements by handling more complex structures, but they still lack the capability to incorporate diverse data types comprehensively. To fully capture the multifaceted nature of climate risks, it is essential to integrate as much data as possible from various sources, including both quantitative and qualitative data, in order to provide valuable insights into insurers' risk assessments.

This need for comprehensive data integration gives rise to the use of foundation models, offering a new opportunity to enhance long-term climate-related analysis in the insurance sector. By integrating foundation models into our analysis, we can leverage multiple sources of data, such as risk assessments and sustainability reports, that can provide us with a more comprehensive understanding of how climate-related risks impact insurance premiums.

This research focuses on the need to strengthen resilience to climate-related hazards, as highlighted in the United Nations Sustainable Development Goal 13.1 [5]. Our proposed approach can enable homeowners and landowners in vulnerable regions to plan for severe climate events and mitigate potential financial risks, thereby enhancing their adaptive capacity. We outline a case study focused on Louisiana, a state particularly susceptible to climate change, where conventional machine learning methods help to provide insights into the impact of climate change on insurance premiums. By expanding on these methods, we aim to create an even more robust framework for predicting insurance premiums, both addressing climate change impacts and highlighting socioeconomic disparities. This approach leverages high-resolution climate downscaling, incorporates multimodal data, and utilizes outputs from pre-trained climate models to improve prediction accuracy. This study bridges the gap in current literature by demonstrating how recent developments in foundation models can significantly improve insurance risk assessments, ultimately supporting more informed decision-making and financial planning in the context of climate change.

## 2 RELATED WORKS

**Machine Learning Bridging Climate Risk and Insurance:** The application of machine learning techniques to insurance and climate change contexts remains understudied. Blier-Wong et al. review the applications of machine learning in actuarial science, particularly

**Unpublished working draft. Not for distribution.**

in ratemaking and reserving, and highlight the recent adoption of advanced algorithms over traditional Generalized Linear Models (GLMs)[6]. Taha et al. propose a framework for improving predictive machine learning techniques in the insurance sector through effective feature selection, which enhances the performance of clustering and classification methods [7]. Sahai et al. evaluate the effectiveness of machine learning techniques in underwriting decisions for life insurance, emphasizing the importance of model interpretability using SHAP values and feature importance[8]. Additionally, Huntingford et al. examine how machine learning can be used to quantify and warn against approaching extreme weather events in climate change scenarios [9]. While these studies use machine learning to improve insurance risk assessment, they do not examine the direct impact of climate risk on insurance premiums, nor do they discuss applications with comprehensive and multimodal datasets relating climate change and insurance.

**Foundation Models For Climate and Risk Prediction:** Foundation models, especially large language models, are increasingly being used in climate and insurance contexts to enhance actuarial tasks and provide detailed risk assessments [10] [11]. In parallel, foundation models have recently been used in weather and climate modeling applications. Ernst explores the use of structured attention transformers to predict weather events, showing improved performance over traditional weather models [12]. Cong et al. leverage both temporal and multi-spectral data to propose a pre-training framework for satellite imagery using masked autoencoders [13]. Yuan et al. propose a transformer-based network to enhance Satellite Image Time Series classification by leveraging unlabeled data [14]. Perhaps the most relevant application of foundation models to climate is ClimaX, a foundation model for weather forecasting and climate projection, which extends the transformer architecture [15]. While these works provide a strong basis for the climate modeling piece of our work, they they do not fully explore integrating multimodal data to study climate forecasts in a risk context, nor do they relate it to financial and socioeconomic contexts.

Our paper aims to bridge these gaps in the literature by integrating foundation models with conventional machine learning techniques to develop a comprehensive risk assessment framework. This framework directly analyzes the relationship between climate change and insurance premiums, highlighting the impacts on demographics and socioeconomic factors.

# 3 LOUISIANA CASE STUDY

Studying insurance trends in Louisiana is crucial due to the state's high vulnerability to climate change, characterized by its geographic location, low-lying topography, and proximity to the Gulf of Mexico. This case study aims to uncover the specific insights that traditional machine learning, particularly using a random forest regression model, can provide regarding the impact of climate change on future insurance rates. By analyzing the following comprehensive datasets, we seek to understand the insights that can be gleaned from traditional machine learning methods before applying foundation models to enhance forward-looking assessments.

## 3.1 Datasets

This analysis utilizes a comprehensive dataset that includes climate data, homeowners' insurance premium data, and demographic and socioeconomic data for each ZIP code in Louisiana.

The climate data includes natural disaster frequency from Arizona State University's Spatial Hazard Events and Losses Database SHELDUS [16], which provides county-level information on natural disaster events. Historial and future (2021-2099) precipitation and temperature data is sourced from CORDEX CORE Downscaled Projections, a framework to provide region-specific climate data that are based on large-scale global climate models [17] [18]. Sea level rise projections are obtained from Scenarios for the National Climate Assessment, the central initiative of the U.S. Global Change Research Program to analyze current and future trends related to global warming [19].

Insurance premium data is sourced from the Quadrant Insurance Premium Data Report and S&P Capital Pro Rate Watch [20], providing average premium prices by ZIP code from 2011 - 2020, for the average household ('Good' FICO credit score and 30 year old dwelling).Socioeconomic data includes average household income from the American Community Survey's Census Database [21], and house price indices from the Federal Housing Finance Agency [22].

The dataset was cleaned, standardized to ZIP code granularity for the years 2011-2020, and consolidated, then split into training and validation sets.

## 3.2 Insurance Premium Forecasting as Supervised Machine Learning

We demonstrate that forecasting insurance premium prices may be modeled as a supervised learning problem. We define the features for model training to be historical insurance premium prices, historical climate data such as average temperature, precipitation levels, and frequency of natural disasters, and demographic information like median household income and population density. Using these features, a Random Forest Regression Model is trained to learn the complex relationships between these variables and predict future insurance premiums.

The final model is a Random Search-based Random Forest, which constructs multiple decision trees during the training process and averages their predictions to forecast future prices. The Random Forest algorithm creates each tree using a bootstrap sample of the data and considers a random subset of features at each split to ensure robust predictions. A Random Search was performed to identify the optimal hyperparameters for the number of trees ($v$) and the maximum depth of each tree ($\delta$). The best-performing set of hyperparameters was selected based on the model's $R^2$ and out-of-bag (OOB) error. Thus, the prediction of the model for a given input $x$ is given by

$$\hat{y} = \frac{1}{v} \sum_{i=1}^{v} T_i(x; \delta)$$

The OOB score is computed as the mean prediction error on each training sample, using only the trees that did not have the sample in their bootstrap sample. After building and training the robust tree-based model, the model was applied to future climate

scenarios, and was used to predict how insurance premiums will change as a result of climate change in Louisiana. The results are outlined below.

### 3.3 Model Results

*3.3.1 Model Evaluation.* The optimal hyperparameters for the model and the average $R^2$ value and Out-of-bag Error on the training data over 50 iterations are displayed in Table 1.

**Table 1: Hyperparemeters and Evaluation Metrics**

| Hyperparameter Tuning | |
| --- | --- |
| Optimal $v$ (hyperparameter 1) | 287 |
| Optimal $\delta$ (hyperparameter 2) | 21 |
| **Evaluation** | |
| Average $R^2$ | 0.2819 |
| Average Out-Of-Bag (OOB) Score | 0.8223 |

The $R^2$ value of 0.28 suggests that some significant degree of the variance in insurance prices can be explained by climatic factors. The OOB score of approximately 0.82 represents a positive generalizability of this model. Given this, there is a strong basis for confidence in the model's ability to accurately predict future trends in insurance premium pricing influenced by climate change. The following results are the outputs of this model.

*3.3.2 Premium Predictions.* The top ten ZIP codes in Louisiana projected to experience the most significant increase in insurance premium prices over the next 50 years have an average premium price of approximately \$6,441.42, which is nearly \$4,000 higher than the projected national average. Figure 1 provides a detailed breakdown of percent changes in insurance premiums by ZIP code in the next 70 years. The visualization illustrates that coastal ZIP codes, as well as those in close proximity to the Mississippi River, are confronted with a significantly higher risk of insurance premium spikes over time. ZIP codes on the southernmost, coastal tip, as well as in the New Orleans area, will face the highest percent change of premium price, surpassing 120% in some cases. This is logical, given the anticipated rise in precipitation-based and sea level-rise-based damages in these areas. Conversely, the northern and northwestern regions of the state will be relatively resilient, with the least expected increase in homeowners' premium costs.

*3.3.3 Premiums as a Function of Household Income and Race.* The analysis reveals that approximately 6% of Louisiana's population will face home insurance premiums exceeding 10% of their annual income, with a significant portion spending over 20%, as shown in Figure 2. Coastal and densely populated areas near the Mississippi River and the Gulf of Mexico are particularly affected, as illustrated in Figure 1. These regions are more susceptible to climate-related risks such as hurricanes, flooding, and sea level rise, which drive up insurance costs.

Furthermore, a demographic analysis of the 50 ZIP codes with the highest projected premium changes reveals that these areas, predominantly inhabited by Black and African American communities, face significant racial disparities, as depicted in Figure 3. These communities are disproportionately impacted by rising insurance costs

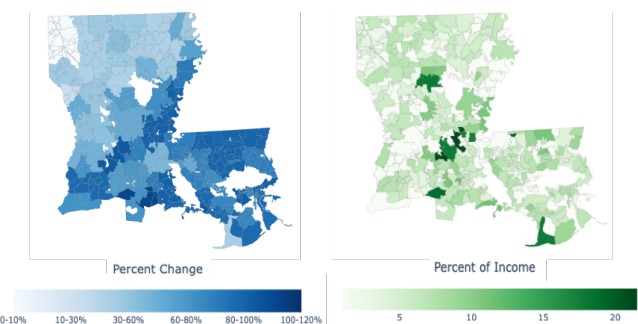

**Figure 1: By 2099: (Left) ZIP codes projected to face the highest insurance premium increase. (Right) ZIP codes projected to pay the highest percent of the average household income towards insurance**

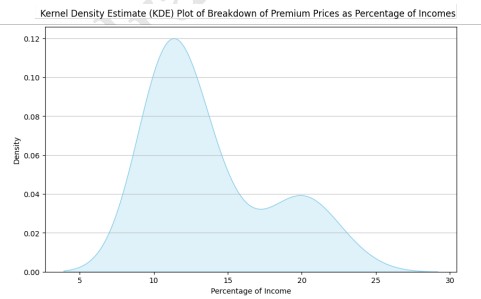

**Figure 2: Distribution of projected premium prices as a percent of income, for those projected to pay 10% or more of their income.**

due to pre-existing socioeconomic vulnerabilities, including lower average incomes and higher rates of homeownership in high-risk areas [23].

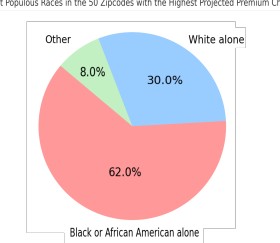

**Figure 3: The racial breakdown of ZIP codes projected to face the highest increases in premium costs**

### 3.4 Limitations of Conventional ML

While our study employing conventional machine learning methods provides significant insights into the dynamics of Louisiana's homeowners insurance market under the influence of climate change, there are notable limitations inherent to these approaches.

Firstly, existing machine learning methods do not fully account for, or quantify, the multifaceted effects of climate risk. Traditional

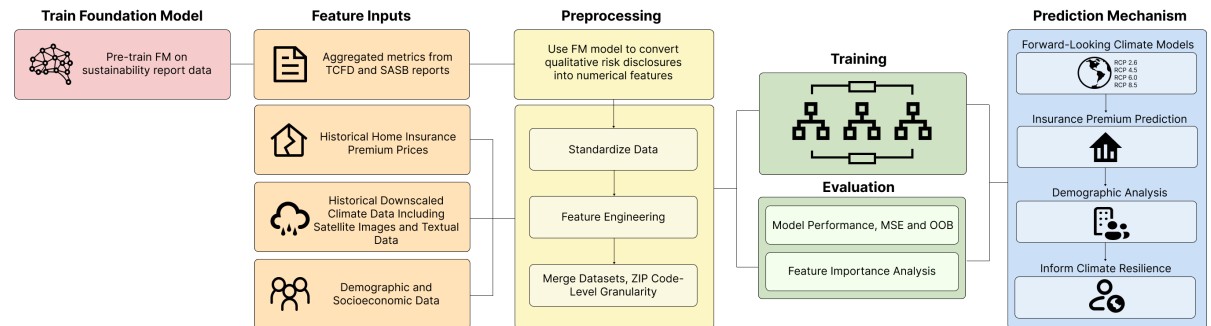

**Figure 4: Proposed architecture for prediction architecture incorporating foundation models.**

models primarily rely on historical data and often lack the ability to integrate diverse climate factors which reflect the full scope of future climate scenarios. As a result, the predictive power of these models is constrained, potentially underestimating the severity of climate impacts on insurance premiums.

Secondly, traditional approaches fall short in that they fail to capture the underlying risk assessment processes employed by insurance companies. Insurers incorporate a wide range of risk assessments that influence insurance rate adjustments and market decisions, and conventional prediction models do not incorporate these mechanisms.

Thirdly, current methods do not adequately address the spatial and temporal variance of climate change. Climate impacts are highly localized and evolve over time, affecting different regions and periods in varying degrees. Traditional actuarial and machine learning models often lack the granularity needed to account for this variance, leading to generalized predictions that do not reflect the localized nature of climate risks.

## 4 ADVANCES WITH FOUNDATION MODELS

To address the limitations of traditional machine learning methods in assessing climate risks' effect on insurance, we propose incorporating foundation models into the risk assessment and price prediction architecture in three main ways, visualized in Figure 4:

*4.0.1 Foundation Models for Climate Downscaling.* Expanding on the work of Nguyen et al, generative foundation models have the capability to produce highly localized climate projections, which are essential for accurately assessing regional climate risks and their effects on insurance. These models can generate high-resolution climate data specific to different geographic areas, potentially accounting for the significant variability in climate impacts within a single state, as seen in Louisiana. By leveraging generative foundation models in the prediction architecture, we can obtain detailed, downscaled climate factors, allowing for a more precise evaluation of spatial variability in insurance premiums.

*4.0.2 Incorporating Multimodal Data.* To align our predictions with the true assessment mechanisms of insurance companies, we propose incorporating multimodal data into predictive foundation models. This includes incorporating aggregated metrics from TCFD (Task Force on Climate-related Financial Disclosures) and SASB (Sustainability Accounting Standards Board) reports, which provide

disclosures on an insurance company's governance and risk management related to climate change. By using foundation models to convert these qualitative disclosures into quantitative features, we can better capture the complex factors that influence insurance rate adjustments.

*4.0.3 Utilizing Outputs of Pre-Trained Climate Models.* Traditional models often rely on historical climate data, which can misrepresent the evolving nature of climate risks. Instead, we propose using the explicit outputs of pre-trained climate models as input features in risk assessment and price prediction. This allows us to incorporate forward-looking climate projections directly into our predictive models, ensuring that our assessments are aligned with the latest scientific understanding of future climate scenarios.

## 5 CONCLUSION

This study examines the impact of climate change on home insurance rates in Louisiana, using a Random Forest Regression model to highlight significant geographic, socioeconomic, and racial disparities. The model predicts premium increases exceeding $4,000 above the national average in high-risk areas and reveals that approximately 6% of the population, particularly Black and African American communities, will face premiums exceeding 10% of their annual income.

Our research supports the United Nations Sustainable Development Goal 13.1 by strengthening resilience and adaptive capacity to climate-related hazards. Advanced risk assessments, enabled by foundation models, play a crucial role in empowering communities to prepare for and respond to climate events, promoting equity in access to insurance and helping these communities achieve greater climate resilience. While Louisiana serves as an example of such a community, the broader implication of this research is that foundation models can be used to provide detailed, localized assessments of climate risks, addressing the limitations of traditional methods. In doing so, our proposed approach can be useful to further improve predictive accuracy and support future policy interventions.

## ACKNOWLEDGMENTS

We would like to acknowledge Professor Peter Tufano for his guidance in this work, and the D3 Institute at Harvard for their support.

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
