# OpenReview forum: "The Future of Foundation Models in Predicting Climate-Related Risks in the Insurance Sector: A Case Study in Louisiana"
_KDD.org/2024/Workshop/Fragile_Earth — Fragile Earth ShortPresentation_

### Official Review · Reviewer_HnBk · 2024-07-08
**The Future of Foundation Models in Predicting Climate-Related Risks in the Insurance Sector: A Case Study in Louisiana**

**Rating:** 6
**Confidence:** 5

**Review:**

In this paper, the author proposes incorporating foundation models into climate risk assessment and insurance modeling to gain deeper insights into the impact of climate change on insurance premiums and the disparities in equal access to affordable insurance. This is more of a positional paper and misses presenting analysis or results out of the proposal workflow in figure 4. The author presents the results out of a random forest regression model on the training data and present (a) the premium predictions by 2099 and (b) project the premium prices as a percent of income.

1. How was the figure1 produced? It would have been nice if the author discussed about the way the figure 1 was produced for the benefit of the readers.
2. It would have been nice if author tried some off the shelf foundational models and presented there results.

---

### Official Review · Reviewer_EZ7j · 2024-07-17

**Rating:** 5
**Confidence:** 5

**Review:**

The paper addresses an important topic of the relationship between the house insurance and climate change. However, it does not place the current study within the existing body of knowledge (including spatio-temporal analysis of claim dynamics) and does not adequately discuss the limitations of the proposed approach.

For references on the topic of the house insurance and climate change"

Dey et al., 2024 Multivariate Modeling of Precipitation-Induced Home Insurance Risks Using Data Depth
Haug 2020  Aspects of Climate-Induced Risk in Property Insurance
Lyubchich et al., 2019 Insurance risk assessment in the face of climate change: Integrating data science and statistics
 Where the Home Insurance Meets the Climate Change: Making Sense of Climate Risk, Data Uncertainty and Projections
Lyubchich et al., 2019 Where the Home Insurance Meets the Climate Change: Making Sense of Climate Risk, Data Uncertainty and Projections
Soliman et al. 2015 Evaluating the impact of climate change on dynamics of house insurance claims
Yuvaraj et al. 2021 Topological clustering of multilayer networks (peril maps of climate-induced risk in insurance using multilayer networks)
Scheel et al., 2013 A Bayesian hierarchical model with spatial variable selection: the effect of weather on insurance claims

---

### Official Review · Reviewer_Xyfh · 2024-07-18
**The Future of Foundation Models in Predicting Climate-Related Risks in the Insurance Sector: A Case Study in Louisiana**

**Rating:** 5
**Confidence:** 3

**Review:**

The paper addresses a highly relevant and timely issue. Climate change and its impact on the insurance sector is a critical topic, especially given the increasing frequency and severity of natural disasters. By addressing below areas for improvement and incorporating additional suggestions, the research could contribute to understanding and mitigating climate-related risks in the insurance sector.
1.	The paper indicates that traditional methods fail to account for the multifaceted effects of climate risk. It would be beneficial to include a more detailed discussion on the limitations of current models and how foundation models specifically address these gaps. Comparative analysis with existing models could enhance the paper's contribution.
2.	The paper could be strengthened by discussing the policy implications of its findings. How can the insights gained from incorporating foundation models into insurance risk assessments influence policy decisions? Addressing this question could make the research more actionable.
3.	Outline potential future research directions that can build on the findings of this study. This could include exploring other vulnerable regions, different types of natural disasters, or further advancements in modeling techniques.

---

### Decision · Program_Chairs · 2024-07-24

Accept (Short Presentation)